# Co-Training for Visual Object Recognition Based on Self-Supervised Models Using a Cross-Entropy Regularization

**DOI:** 10.3390/e23040423

**Published:** 2021-04-01

**Authors:** Gabriel Díaz, Billy Peralta, Luis Caro, Orietta Nicolis

**Affiliations:** 1Departamento de Ciencias de Ingeniería, Facultad de Ingeniería, Universidad Andres Bello, Antonio Varas 880, 8370146 Santiago, Chile; g.diazcalderon@uandresbello.edu (G.D.); billy.peralta@unab.cl (B.P.); 2Departamento de Ingeniería Informática, Facultad de Ingeniería, Universidad Católica de Temuco, Rudecindo Ortega 2950, 4781312 Temuco, Chile; lcaro@uct.cl

**Keywords:** co-training, deep learning, semi-supervised learning, self-supervised learning

## Abstract

Automatic recognition of visual objects using a deep learning approach has been successfully applied to multiple areas. However, deep learning techniques require a large amount of labeled data, which is usually expensive to obtain. An alternative is to use semi-supervised models, such as co-training, where multiple complementary views are combined using a small amount of labeled data. A simple way to associate views to visual objects is through the application of a degree of rotation or a type of filter. In this work, we propose a co-training model for visual object recognition using deep neural networks by adding layers of self-supervised neural networks as intermediate inputs to the views, where the views are diversified through the cross-entropy regularization of their outputs. Since the model merges the concepts of co-training and self-supervised learning by considering the differentiation of outputs, we called it Differential Self-Supervised Co-Training (DSSCo-Training). This paper presents some experiments using the DSSCo-Training model to well-known image datasets such as MNIST, CIFAR-100, and SVHN. The results indicate that the proposed model is competitive with the state-of-art models and shows an average relative improvement of 5% in accuracy for several datasets, despite its greater simplicity with respect to more recent approaches.

## 1. Introduction

Automatic visual recognition consists of learning visual categories using a computer to later identify new instances of such categories. The vast majority of computer vision tasks are primarily based on the ability to recognize categories as well as specific objects. This area has gained great development at present due to the variety of potential applications in multiple areas of information, among which are content-based image search, video data mining, or object identification for mobile robots [1]. Visual category recognition seeks to recognize different instances of a generic category that belong to the same conceptual class, for example, people, houses, or cars. The great difficulty in this task is due to the fact that there are multiple variations in appearance between different instances of the same category due to pose, occlusion, deformation, or lighting [2,3]. Currently, notable progress has been made in this area through the use of deep learning algorithms [4].

Deep learning consists of the training of artificial neural networks that are made up of multiple processing layers. Deep learning tasks mainly encompass two types of learning: Supervised and unsupervised learning. Supervised learning [5] is based on creating a model that returns the label of an input data using examples where its respective label is known. For example, in the object classification task, each object image is associated with the label that represents the object’s category. On the other hand, unsupervised learning is based on creating a model that allows finding hidden patterns of the data. In contrast to supervised learning, there are no explicit labels associated with each input. Typical tasks of unsupervised learning are clustering and density estimation [6].

Visual recognition using deep learning techniques typically requires a large number of labeled images due to the huge number of parameters of such neural network models [4,7]. On the other hand, the number of unlabeled images is usually much greater proportionally to the number of labeled images [8]. The required work to label the images needs a human who manually perform the labeling. This process can be costly, as it requires a lot of human time to label these images, as well as being prone to human factor errors. Therefore, a reasonable way to use the information from the unlabeled data is through semi-supervised learning. A detailed review of semi-supervised algorithms can be found in [9]. A popular semi-supervised learning technique is the co-training algorithm where it is assumed that you have information from complementary views; these views allow us to use few labeled data to build a more robust model than the initial one given by the labeled data [10].

The co-training algorithm requires knowing the views of the training images, which are generally not in a visual object database. On the other hand, deep neural networks tend to overfit in the presence of little data, so it can be expected that when training several neural networks, within a co-training algorithm, they cannot generalize correctly at the beginning. Furthermore, they do not necessarily evolve in a complementary way when considering a training based on fine adjustment, that is, where the weights are similar at the beginning. Although there are complex proposals based on adversarial learning or adversarial networks, we believe that a solution based on the original co-training algorithm has the advantage of being algorithmically simple.

In this research, the development of a variant of co-training model based on multiple views and on deep neural networks is proposed. The model uses self-supervised learning algorithms for complementary aspects of objects, such as degree of rotation or type of filter. The use of such aspects allows the views to differ from each other initially. To facilitate the diversity of view models, required by the co-training algorithm [10], we propose to differentiate the outputs of the views by incorporating the maximization of the cross-entropy between them. Cross-entropy can be interpreted as a measure of distance between two probability functions. Therefore, the cross-entropy function acts as an entropic regularizer for the neural network classification cost function of views by allowing the outputs of the views to differ from each other. The idea of entropic regularization has been applied in the context of optimal transport problems [11], Markov Decision Process [12], and data classification [13]. On the other hand, this regularization is facilitated because the cost function is also given by the cross-entropy between the results of the view and the actual results, that is, the cost and regularization functions have similar scales. By achieving different views during the training process, the co-training algorithm enables efficient learning of visual object classifiers by considering a small initial amount of labeled data.

This work is organized as follows: Section 2 shows the theoretical framework as well as a bibliographic review of the applied co-training methods in visual recognition. Section 3 presents and justifies the proposed variant of co-training. Section 4 shows the results of the experiments carried out to validate the proposed model. Finally, conclusions and further developments are presented in Section 5.

## 2. Background

### 2.1. Theoretical Framework


**Self-supervised learning**


Self-supervised learning is based on surrogate tasks that can be formulated using only unsupervised data in such a way that to reach their goal they require learning characteristics and representations of the original object [14]. The advantage of this type of learning is to avoid the costly and time-consuming process of labeling the data. This type of learning has been applied with relative success in tasks of supervised recognition of visual patterns. Interestingly, it has been found that the classification patterns of objects, for example people and cars, are related to the classification patterns of aspects of the object, for example the angle of rotation of the object. On the other hand, this type of learning has several variants in visual recognition such as the learning of the rotation angle [15], gray scale prediction [16]; as well as in natural language recognition tasks [17], among others.

In relation to visual recognition, self-supervised learning is usually based on performing pretext tasks where the characteristics are learned using the cost functions of such pretext tasks, such as the prediction of rotating an image. Note that the angle label generation can be done automatically. In Figure 1, the labels of self-supervised convolutional neural networks *P* for pretext task, also known as pseudolabels, are automatically obtained without human intervention by applying a transformation operation over the original image, i.e., the rotation of an image by a random angle. The optimization consists of minimizing the error between the convolutional network prediction *O* and pseudo-labels *P*; where the network finally manages to learn visual characteristics of the images.

Multiple self-supervised learning methods have been proposed in recent years. In relation to visual object recognition, we can mention some works. Some self-supervised learning methods are based on image colorization [16], image inpainting [18], Image Jigsaw Puzzle [19], geometric transformation [20], or clustering [21]. A most detailed revision of the current self-supervised methods for visual recognition can be found in [22]. Although there are multiple sophisticated self-supervised methods, we must emphasize that the proposed method is focused on semi-supervised learning where the goal is bringing diversity to the training set. We must remember that the co-training algorithm requires that the views be complementary. The use of self-supervised networks is exclusively to give the complementary information for each training image. We only consider two basic self-supervised methods based on rotation and filtering tasks because these tasks are semantically distant. We think that the semantic distance of both tasks are enough for our proposal. Our intuition is that the features generated by these self-supervised networks can bring diversity to the view’s networks.


**Co-training**


Successful cases of modern machine learning are based on the use of labeled data during model training. These supervised learning techniques generally use data that has been labeled by human experts; however, this procedure is time expensive because it typically requires an expert to label the data. On the other hand, there is often much more unlabeled available data than labeled, which implies that there is information that could be useful. An option for the joint use of labeled data as well as unlabeled is through the application of semi-supervised learning, to which the co-training algorithm belongs. This technique simultaneously uses labeled and unlabeled data to increase the performance of a learning algorithm by assuming that you have access to different views of the data and typically there is a small set of labeled data [4].

The co-training method for the automatic classification task uses unlabeled data to improve learning precision in this problem, where new data can be labeled using different views, which correspond to characteristics that allow describing an aspect of an image. In particular, it is required that each view is sufficient to label the data better than by chance, and that they are complementary to each other. The simplified process of co-training is shown in Figure 2. While the labeled images $X_{L}$ are used to train the classifiers of the views $C_{1}$ and $C_{2}$, they are then applied on the unlabeled images $X_{U}$. Considering the labeled images with greater confidence, each view independently feeds the set of labeled images $X_{L}$. This procedure continues until all the data is labeled.

### 2.2. Related Works

The use of unlabeled data to improve models that require labeled data corresponds to a semi-supervised learning problem. The main problem is that the labeling process can be expensive, even though it is possible to use non-expert taggers [23], and it is more efficient to consider the large volume of unlabeled data as a source of information. Within semi-supervised learning, there have been recent works for its application in the context of deep learning, where some proposals based on adversarial generative networks (GAN) given in [24,25,26] stand out. These networks adapt themselves to use the information available from the data without labels by artificially increasing the labeled data considering the distribution of the unlabeled data, so that a more robust data set is obtained for the improvement of the performance of the classifiers. Other alternatives are based on semi-supervised variants such as co-teaching [8], where they propose the use of two co-operative neural networks to improve the training of data with extremely noisy labels.

On the other hand, the co-training algorithm was initially proposed for the classification of web pages [10] generating views with different inputs and separating the data set into two views, the text and the hyperlinks, which contain information complementary to each other, as well as in sentiment analysis [27], among other applications. Recently, multiple works have emerged that apply deep neural networks considering the co-training algorithm. In [28], it is proposed to create bags of instances considering the distribution of the existing classes, thereby feeding the deep learning algorithm. A relevant proposal in this area is the deep co-training algorithm [29], which focuses on the differentiation between two or more views through the injection of adversarial data that minimize cost that considers the performance of classifiers by view as well as the diversity of the outputs.

The proposals based on the classic semi-supervised learning algorithms [8,28] are generally simple to implement, however, they do not necessarily obtain the best performance. In particular, co-teaching [8] is based on the idea that two networks teach each other each mini-batch considering data with estimated clean labels. This technique has some similarity with our proposal, however, our results are better. We believe that this technique is oriented towards combating data with noisy labels that may be different from our goal. In the case of mean-teacher [30], this method considers a convolutional neural network, while our method considers a ResNet. This could represent a disadvantage for this method, however, we had computational limitations to test multiple base classifiers. For this reason, we chose ResNet as our base classifier in a similar way to [29]. On the other hand, works based on adversarial networks and adversarial generator algorithms [23,24,25,26,31] generally have high performance. However, they entail the complexity of the system when generating new data either through the GAN network, which is known to have convergence problems, as well as those based on adversarial algorithms that depend on parameters different from the original task. In our case, this proposal does not generate new data, which is why it is much simpler to implement than the alternatives shown, and also the performance is reinforced when considering the generation of adversarial patterns between the views generated, meaning that the models will perform a complementary learning to seek such differentiation. This proposal benefits the use of information available without the cost of a label given by the outputs of neural networks based on self-supervised learning, which can be complementary in a natural way.

There are several relevant aspects of this work that we want to discuss. In the case of classic co-training algorithm, we might think that the training could have been affected by the neuronal co-adaptation [32]. Co-adaptation consists of a high dependence between neurons. This could be problematic because if an independent neuron receives noise inputs then their dependent neurons can learn the same wrong pattern. We might think that both view networks could have co-adaptation problems, however, this problem is related to the training of a single network. In our proposal, each view network considers classic techniques to combat co-adaptation as drop-out, therefore, this problem does not seem especially specific for this work. In relation to the use of complementary data, we compare this with data augmentation methods [33]. Data augmentation methods, in the context of visual object recognition, typically consist of increasing the number of training images by adding slightly modified copies of existing training images. Nonetheless, this technique does not correspond to complementary views since data augmentation typically follows a random process, e.g., a random crop or image resizing. In the case of our method, each view is clearly associated to semantic features given by the rotation and filter features. A more sophisticated technique for data augmentation is given by the AutoAugment algorithm [34]. This technique consists of a simple search of augmentation policies composed by a set of image processing operations as translation, rotation, or shear. The selection of these policies is guided by performance in a validation set. Although this technique is powerful, it is related to a supervised setting. In fact, the experiments reported in the original paper consider the entire labeled datasets. In our case, we focus on semi-supervised learning where we assume a small amount of labeled data. Furthermore, we look for using a variant of the co-training algorithm where the views must be complementary.

## 3. Proposed Method

Our proposal is based on the idea that self-supervised learning algorithms can generate views that complement each other, without requiring the use of labels in the data. Regarding complementary, we can see that the self-supervised variant based on the pretext task of predicting image rotation is quite a different task from predicting the degree of filtering of an image, which suggests that these tasks can bring diversity to the co-training algorithm. Based on this idea, we propose a co-training model based on classifiers given by deep neural networks to the views of the main algorithm. In our work, these views correspond to object recognizers based on pre-trained deep convolutional neural networks. These views incorporate supplementary information given by the output of an upper layer of a self-supervised neural network which will reinforce the complementary of the views, according to the initial idea. On the other hand, deep networks tend to fit well to the data when starting from an initial training given by massive databases; therefore it is feasible that the outputs of the networks tend to resemble each other as the co-training algorithm advances. Therefore, in training we additionally consider a view differentiation cost; in other words, we want the views to differ while simultaneously seeking to minimize the difference with the actual labels of the images. Since this proposal merges the concepts of co-training and self-supervised learning considering the differentiation of outputs, we call this algorithm Differential Self-Supervised Co-Training (DSSCo-Training).

We should note that the self-supervised and semi-supervised components share the use of unlabeled data. Nonetheless, both methods are essentially different. While self-supervised learning does not require the original labels, semi-supervised learning requires a small number of labeled samples. In relation to our proposal, the focus is on semi-supervised learning given by the proposed variant of the co-training algorithm. Self-supervised networks are used as complementary networks that help improving the performance of DSSCo-Training and are trained once in the initial step of our method which corresponds to line 5 of Algorithm 1.

Our method seeks to generate a reliable classifier considering a small proportion of labeled images compared to unlabeled images. We think that the set of unlabeled images can improve the performance of the initial model based on training with the labeled data. Our algorithm considers the incorporation of a selection process and addition of unlabeled images. While our algorithm requires the classification agreement between the views as a requirement for the addition of the unlabeled images, the classic co-training algorithm adds unlabeled images considering each view independently. For example, when considering views 1 and 2, if the view classifier 1 determines that the label of an image is 1 and the view classifier 2 determines that that same image has a label 0, then the labeled image is considered unreliable, so it will not be added to the training.
**Algorithm 1:** Proposed model training algorithm.
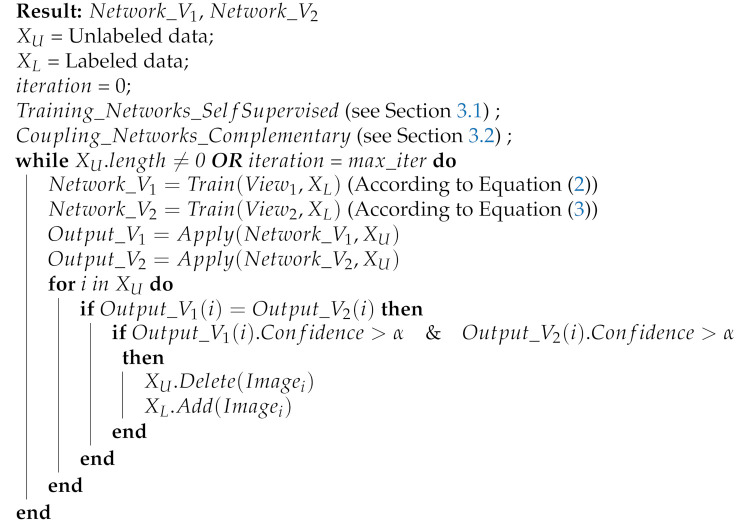


Algorithm 1 shows the complete process of the proposed model considering the use of two views, which can be extended to more. It starts with training the neural networks of the two views ($Network\_V_{1}$ and $Network\_V_{2}$) using the data labeled $X_{L}$ according to the proposed optimization (see details in Section 3.3). Then, it generates the sort prediction by view using the unlabeled data $X_{U}$ getting $Output\_V_{1}$ and $Output\_V_{2}$. These outputs are compared with each other, and if they are equal and have a high level of accuracy α (in our case we use 0.9), then we use them as new labeled data. In case an image does not meet the previous condition, this image will not be labeled in the current iteration, therefore it will remain in the $X_{U}$ set. We may note that both views networks use the same labels because the labeled set $X_{L}$ is common for both networks. This process is repeated until there is no data left to label or the maximum number of iterations max_iter is reached. If the number of iterations reaches the value of the variable max_iter, the data from the unlabeled set $X_{U}$ will not be used to train the neural networks of the views.

After the execution of Algorithm 1, the result will be two independent neural networks trained with the same data set. During the network testing phase, only one network of the two available views will be used. In the experiments, we observed that the networks already trained have quite similar performance, which is also observed within the co-training framework in [29]. For this reason, we choose the same network for all experiments. In particular, we always used the rotation view neural network during testing.

### 3.1. Self-Supervised Neural Network Training

The views initially correspond to the supervised model that considers the labeled data, this model refers to the base neural network; they consider complementary information that allows them to generate differential training. Each original view will be coupled to a complementary neural network through the penultimate layer of the latter. While one network corresponds to the supervised model of the original view, the other complementary network corresponds to a predefined self-supervised model. That is, the prediction of the labeled data will take into its training the characteristics given by both the supervised and self-supervised models.

Two self-supervised models will be generated, the first one will be for predicting the degree of image rotation and the second will be for predicting the type of filters applied to images. These models are initially trained on all the images in the databases considering standard convolutional neural networks. Then, these are used as complementary networks in the proposed co-training model. For the training of the predictor model of the degree of rotation, the inputs will be the total set of images (labeled and not labeled), to which a label will be associated with its respective rotation given randomly (with values: 90, 180, 270, and 360 degrees). This network can be visualized in Figure 3.

For the training of the filter type model, the inputs will again be the total set of images, to which a label will be assigned that will indicate the type of filter applied randomly (averaged, blurred, Gaussian, and based on median). These pretext tasks were selected for the experiments to reinforce the complementary information between the views.

### 3.2. Coupling Complementary Networks to Original Views

After the previous step, the outputs of the penultimate layer of the self-supervised neural network are added to the penultimate layer of the supervised model network; that is, the outputs of the self-supervised model are the intermediate input of the supervised model. It should be noted that the weights of the self-supervised networks do not change during the training of the main algorithm. In Figure 4, the coupling of the original view 1 with the complementary characteristics of the semi-supervised filter model is shown, where the output of the penultimate layer of the filter model is extracted. It is then resized and applied as an input that connects to the final layers of the original network of the view 1.

The procedure shown in Figure 4 will be applied in a similar way to view 2 of this semi-supervised model, that is, a view contemplates a model that receives as an intermediate parameter the output of the penultimate layer of a self-supervised model considering the rotation and filter cases according to the training view. Finally, the goal of each of the views will be the prediction of the label of the labeled data set using the neural object classification network.

### 3.3. Model Optimization

The proposed model is based on the co-training algorithm, where each training image of the unlabeled dataset is added to the labeled dataset if both views agree with high confidence. This is considered using a threshold α, which is selected experimentally. With the new labeled dataset, the neural network of each view is optimized.

Assuming that in each iteration we have a set D={X,Y} where *D* represents the data labeled in an arbitrary iteration and which is formed by the input data $X=\{x_{1},\ldots ,x_{N}\}$ and output $Y=\{y_{1},\ldots ,y_{N}\}$. Assuming two-view optimization, the total cost function is given by:(1)$$\begin{array}{cc} E_{T} & =E_{V}^{1}+E_{V}^{2}+\beta E_{D}\\ \; & =-\sum_{i=1}^{N}y_{i}log\left(h_{1}\left(x_{i}\right)\right)-\sum_{i=1}^{N}y_{i}log\left(h_{2}\left(x_{i}\right)\right)+\beta \sum_{i=1}^{N}h_{2}\left(x_{i}\right)log\left(h_{1}\left(x_{i}\right)\right) \end{array}$$

The classification cost function for view 1 is given by $E_{V}^{1}$, while the corresponding one for view 2 is given by $E_{V}^{2}$. On the other hand, the success of the co-training algorithm requires that the views be complementary. For such reason, we add the differentiation function $E_{D}$ between views 1 and 2, which is graduated according to the hyperparameter β, which is estimated experimentally. The function $E_{D}$ regularizes the cost classification functions $E_{V}^{1}$ and $E_{V}^{2}$ because it constrains their optimizations incorporating the diversity of the models. Considering the view 1 and the instance *i*, the classification function corresponds to the cross-entropy of the output of the network of the view 1, $h_{1}\left(x_{i}\right)$, with respect to the true output $y_{i}$. The same function is considered for view 2. In the case of the differentiation function $E_{D}$, since instead of minimizing the difference it sought to maximize it, it corresponds to the negative of the cross-entropy of the output of view 1 with respect to the output of view 2. Therefore, the differentiation function $E_{D}$ corresponds to a cross-entropy regularization function. In relation to parameter learning of the model, when we optimize the parameters of view 2, we freeze the values of differentiation function $E_{D}$. The resulting cost function for both views are given by: (2)$$\begin{array}{c} \begin{array}{cc} E_{T}^{1} & =-\sum_{i=1}^{N}y_{i}log\left(h_{1}\left(x_{i}\right)\right)+\beta \sum_{i=1}^{N}h_{2}\left(x_{i}\right)log\left(h_{1}\left(x_{i}\right)\right) \end{array} \end{array}$$(3)$$\begin{array}{c} \begin{array}{cc} E_{T}^{2} & =-\sum_{i=1}^{N}y_{i}log\left(h_{2}\left(x_{i}\right)\right) \end{array} \end{array}$$

These equations imply that we only differentiate the outputs of view 1 with respect to the outputs of view 2, which simplifies the code. The experiments evidence that the proposed solution is able to differentiate between views. Note that this model can be directly extended to the use of more views following the scheme of [29].

## 4. Experimental Results

In this section, the experiments carried out to test the proposed model, DSSCo-Training, are described by specifying the datasets, the configurations, as well as the results in comparison with other similar models. The basic neural network model associated to each view and which performs the task of visual recognition is the Resnet-50 [35]. The Resnet architecture has been used successfully in various object recognition tasks and is based on residual processing layers that are sequentially coupled. In particular, our hardware consists of an Intel Xeon machine with 16GB RAM and Tesla V100 GPU. Due to the balance between performance and available hardware, we consider a number of layers for the Resnet equal to 50.

The proposed algorithm is applied to four standard databases of visual objects. Each one is detailed below.
**MNIST**: This dataset presents a total of 70,000 images of handwritten digits divided evenly into 10 categories [36].**CIFAR-10**: This dataset presents a total of 60,000 images divided evenly into 10 categories. Examples of categories are: airplane, car, dog, boat, etc. [37].**CIFAR-100**: This dataset presents a total of 60,000 images divided equally into 100 categories. Examples of categories are: Dinosaur, oak, tank, raccoon, etc. [37].**SVHN**: This dataset presents a total of 600,000 (99,289 assigned to the training and 531,131 as additional samples) images of the house digits in Google Street images, evenly divided into 10 categories [38].

For the experiments, we followed the scheme proposed in [29] to partition the training and test data. For the MNIST database, we used the 20% of the total data (12,000 images) as the labeled set while the rest of images was used as the unlabeled set (48,000 images). For the CIFAR-10 and CIFAR-100 databases, 4000 labeled images were used corresponding to 6.6% of the total data, the rest of the images (46,000) were used as unlabeled data set in support of unsupervised training of the proposed algorithm. A total of 1000 labeled data was used for the SVHN database and the rest (72,257) as unlabeled data. For ending the proposed algorithm, two term criteria were considered: The ending of the complete labeling of the unlabeled data and the number of iterations to perform, that is, the number of times that update the networks of each view.

The algorithm parameter settings are shown in Table 1. The number of iterations is considered as the number of times the algorithm is executed, which is a term criterion of the algorithm. The number of epochs corresponds to the number of iterations for training the neural network model. Finally, the error percentage was used as an evaluation metric, which consists of the number of misclassified images divided with the total number of images.

The choice of the configuration of the parameters, number of iterations, number of epochs, and β of the proposed model was based on several experiments with the training partition of CIFAR-10 dataset considering a hold-out scheme. In particular, we tried the combinations of the values of the sets {5,10}, {20,50,100}, and {0.001,0.01,0.1}, respectively, for number of iterations, number of epochs, and the regularization parameter β. The same hyperparameters were considered for all datasets. These values were not optimized for each dataset because we wanted to obtain a fair comparison of our method with respect to the competing methods, however, our results indicated that the chosen values are acceptable. Nonetheless, it is possible to improve the resulting metrics considering a careful selection of hyperparameters for each database, as can be seen in the sensibility analysis given in Section 4.1.2. Table 1 shows the values of the main parameters of the model that were used in all the performed experiments, excluding the sensitivity analysis.

The experiments carried out are based on the quantitative and qualitative analysis of the proposed model, DSSCo-Training. In the quantitative analysis, we will first compare the performance of DSSCo-Training with the classic co-training algorithm, as well as with state-of-the-art techniques of semi-supervised learning based on classic variants [8,26,30], in adversarial generative networks [24,25], and using adversarial instances [29]. In addition, we will perform a sensitivity analysis on CIFAR-10 to study the effect of the regularization factor based on cross-entropy and the number of internal iterations. Finally, a qualitative analysis will be carried out where we will visually review the instances used by the DSSCo-Training through the iterations of the external loop of DSSCo-Training given in Algorithm 1.

The experiments and the construction of the models were carried out using the Python programming language, where the Keras library specialized in deep networks was used. This library, which is based on TensorFlow, contains the implementation of multiple models that have accelerated the implementation of this proposal. In the case of state-of-the-art algorithms, we consider the results published for each of them. The demo source code for this work can be found at https://github.com/Anonim1121/DSSCoTraining (accessed on 27 March 2021).

### 4.1. Quantitative Analysis

#### 4.1.1. Classification Performance

Table 2 shows the results of the application of the proposed algorithm, making a comparison with other state-of-the-art proposals considering the databases *CIFAR-100* and *SVHN*.

Table 3 shows the results obtained with the CIFAR-100 and MNIST databases. The results of the experiments are compared with those of other works. For a better visualization, we report these datasets in a separate table given the available results of the compared techniques.

Regarding the databases *CIFAR-10* and *SVHN*, the obtained results show that the DSSCo-Training is superior to the other methods with the exception of the Deep Co-training method. With regards to the databases *CIFAR-100* and *MNIST*, the results obtained through the proposed method, DSSCo-Training, show a superiority in terms of the classification error in relation to the models found in the state of the art. In particular, as shown in Table 3, with respect to *CIFAR-100*, our method presents an relative improvement of 4.02% compared to deep co-training and 2.41% respect to co-teaching. In the case of *MNIST*, our method obtains a relative improvement of 6.01% over co-teaching.

In relation to the comparison with deep co-training, we observe that while this algorithm uses an optimization process to generate new data in addition to those of the original training set, the proposed model does not make any alteration to the data and is not dependent on an optimization external to co-training as required by the generation of adversarial data. Furthermore, the use of the general scheme of the co-training algorithm considerably simplifies the implementation of our model.

#### 4.1.2. Sensibility Analysis

This section shows the sensitivity analysis of the performance of DSSCo-Training with respect to the value of the regularization hyperparameter β and the number of epochs used in the training of the neural networks of the views. The number of epochs is included in this analysis because it experimentally influences the performance of the model. This is explained because the small number of data tagged at the beginning facilitates the overtraining of neural networks. In these experiments for the value of the hyperparameter β the values [0.001,0.01,…,0.09,0.11,…,0.16,0.1,…,0.9] are tested, while the numbers of epochs tested correspond to [20,40.60,100].

Table 4 shows the precision obtained in CIFAR-10 by varying the value of β and the number of epochs. This experiment is only performed on this database due to the high computational cost required.

The results of Table 4 show that if the value of β is very large, the error percentage tends to be too. It can also be seen that the best results correspond to a value of β between 0.01 and 0.05. We experimentally observe that a value of β greater than or equal to 0.8 causes the algorithm to learn a model that does not converge during the training process. We suggest that this is because the difference between view models is over-prioritized. Because of this, only β is reported up to 0.7. We observe that cross-entropy-based regularization requires a small weight with respect to the classification cost function, as seen in the best value of β, that is, 0.02. However, a not very small value of β appears decisive in the performance obtained, as can be observed when experimenting with β equal to 0.001. Note that the value of β equal to 0 is equivalent to the classic co-training algorithm, whose results are shown in Table 2 and Table 3. We believe that the poor results of the classical training algorithm are due to the fact that the deep neural networks of the views tend to give similar results to each other during training, which hinders the usefulness of the co-training algorithm. Furthermore, we hypothesize that this collapse of view models, in terms of their outputs, is explained because the two view networks are not complementary in contrast to our method. This collapse of models was also observed in [29]. This hypothesis is supported by the fact that in all the experiments our proposal improves the classic co-training algorithm. In this work, we focus on the comparison of our method with respect to classical co-training and several state-of-art methods, however, an interesting future topic is the study of the view representations, e.g., how different are the resulting views considering the iterations of the main algorithm? Overall, this analysis suggests that DSSCo-Training performance can be improved by making a careful choice of the regularization hyperparameter, as well as the number of internal iterations of training views.

### 4.2. Qualitative Analysis

In order to visualize the unlabeled images incorporated into the training set in each of the iterations and to qualitatively analyze the capacity of the presented model to incorporate more complex characteristics in each one of them, a random sample of these images was extracted in the truck category of dataset CIFAR-10. As shown in Figure 5 for each iteration, a selection of images with different and more complex characteristics can be seen. This suggests that the model is able to recognize more difficult objects after each iteration of the co-training algorithm. For example, in iteration 1, the front of the trucks is well delimited; while in iteration 3, the trucks are more mixed with surroundings, as in the last example where the boundary of the black truck appears mixed with the background.

## 5. Conclusions

This work introduces DSSCo-Training, a semi-supervised learning method for visual object recognition based on the co-training algorithm using deep neural networks supported by self-supervised auxiliary networks considering rotation and filter models. In particular, the high improvement obtained over the classic co-training model is related to the use of regularization based on the cross-entropy of the outputs of the neural models of the views. The experiments consider real databases of visual objects of various characteristics. The results show that the proposed technique is competitive with state-of-the-art techniques of semi-supervised learning in several databases, or even outperform them in some cases. We believe that it is relevant to highlight that this proposal only uses self-supervised patterns already known from the unsupervised data, that is, it does not add higher computational cost during the training of the co-training algorithm. On the other hand, this proposal uses a version of the co-training algorithm without resorting to extra optimizations as in more complex models. This work provides evidence, for the first time, of the utility of self-supervised neural networks in the co-training algorithm.

As future work, we plan to improve the model by incorporating joint optimization of the co-training models simultaneously with the self-supervised neural models of the auxiliary classifiers. Another possible route is the exploration of more sophisticated self-supervised models than those used in this work. Finally, we hope to investigate the use of adversarial models to further improve the classification performance.

## Figures and Tables

**Figure 1 entropy-23-00423-f001:**
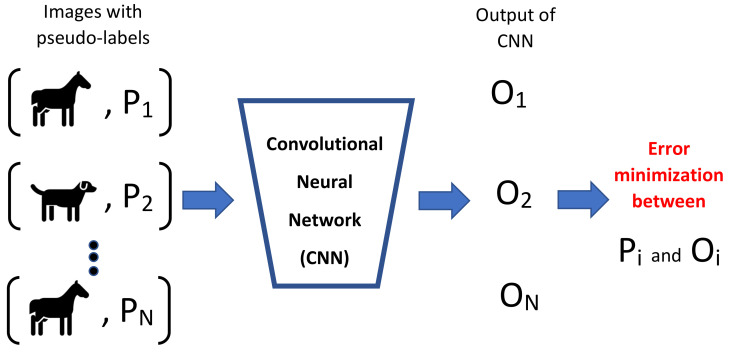
Typical self-supervised learning flow in visual recognition. A network learns pretext tasks whose labels can be generated automatically.

**Figure 2 entropy-23-00423-f002:**
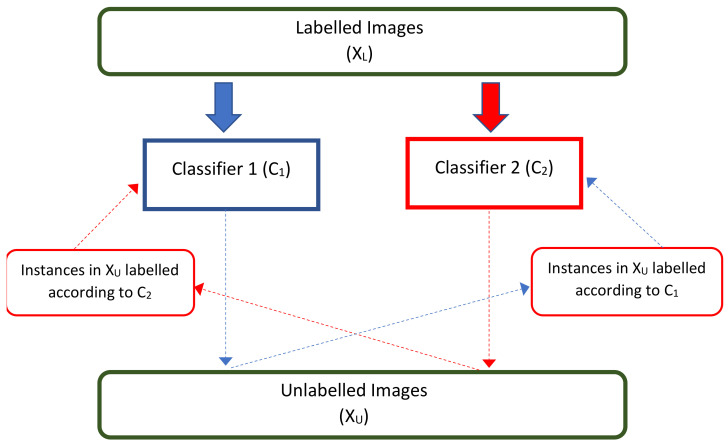
Co-training process. View classifiers complement each other, where the output of one classifier feeds the other.

**Figure 3 entropy-23-00423-f003:**
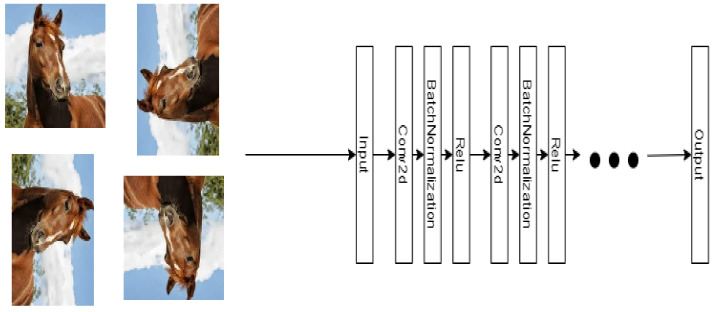
Self-supervised rotation model is represented with a convolutional neural network. In the example, the model seeks to predict the angle of rotation of the object horse. The self-supervised filter type model also corresponds to a convolutional neural network.

**Figure 4 entropy-23-00423-f004:**
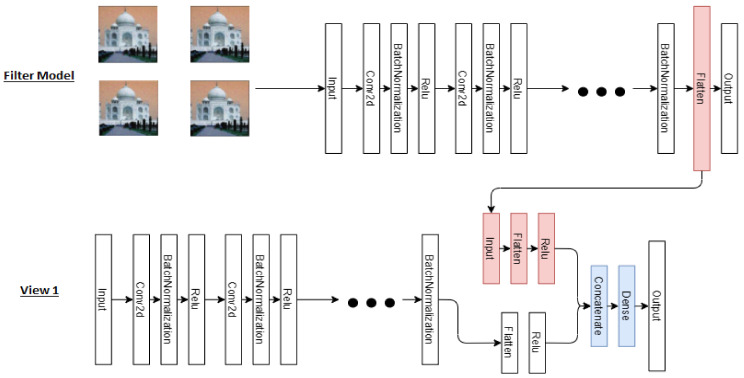
Representation of view 1 considering the self-supervised filter model. This structure is replicated for view 2 which considers the rotation model. The self-supervised filter network predicts the applied filter. The view network predicts the object class. These two networks are connected by a link between a Flatten layer of the Filter network and an Input layer of the View network. The output function of the Flatten layer corresponds to a vectorization of a normalized convolutional operation.

**Figure 5 entropy-23-00423-f005:**
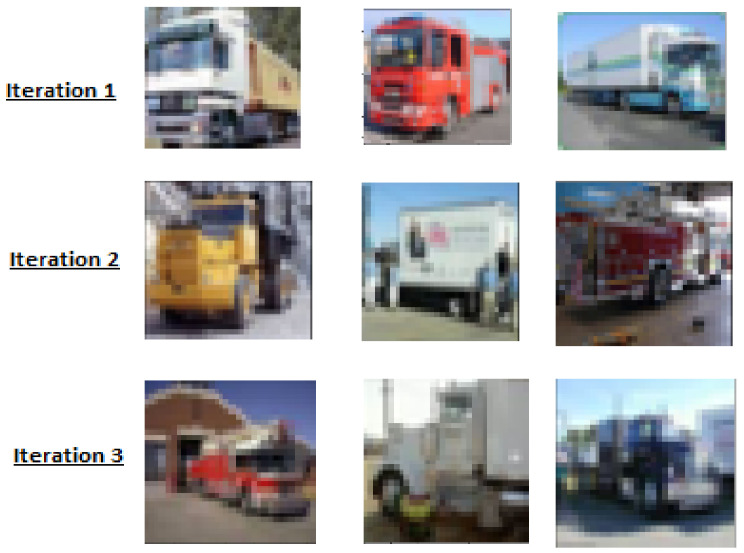
Images selected in iterations 1, 2, and 3. As the iterations in these examples increase, samples are more difficult to recognize.

**Table 1 entropy-23-00423-t001:** Parameters of the proposed model, DSSCo-Training.

Field	Value
Optimizer	Adam
Loss	Equation (Equation 1)
Iterations	5
Epochs	50
β (from Equation (Equation 1))	0.01

**Table 2 entropy-23-00423-t002:** Experiments using the databases *CIFAR-10* and *SVHN*, taking the error percentage as evaluation metric. The best results are indicated in bold and the second best in italic.

Algorithm	CIFAR-10	SVHN
GAN [25]	18.63	8.11
Deep Co-Training [29]	**9.03**	**3.61**
Co-Teaching [8]	15.45	—-
Bad GAN [39]	14.41	4.25
Temporal Ensemble [26]	12.16	4.42
Mean Teacher [30]	12.31	3.95
DSSCo-Training	*11.32*	*3.85*
Classical Co-Training	44.01	22.03

**Table 3 entropy-23-00423-t003:** Experiments using the databases *CIFAR-100* and *MNIST*, taking the error percentage as evaluation metric. We only indicate the best results with bold.

Algoritmo	CIFAR-100	MNIST
Deep Co-Training	38.77	—–
Co-Teaching	38.13	8.08
DSSCo-Training	**37.21**	**7.10**
Classical Co-Training	61.02	34.86

**Table 4 entropy-23-00423-t004:** Sensitivity analysis of the error with respect to the hyperparameter β and the number of iterations used in the training of the view models considering the database *CIFAR-10*. The best accuracy value is in bold.

β/Epoch	20	40	60	100
0.001	20.23	22.11	21.82	29.03
0.01	12.94	11.32	11.38	18.23
0.02	12.02	**10.44**	11.36	17.17
0.03	12.78	11.39	11.38	15.95
0.04	13.15	11.44	11.40	12.33
0.05	16.36	16.10	15.87	15.88
0.06	19.13	18.01	18.29	20.44
0.07	19.15	19.23	19.55	23.04
0.08	20.59	21.65	21.90	25.70
0.09	23.44	22.63	22.06	25.33
0.11	24.73	23.24	22.09	25.07
0.12	24.81	23.31	22.16	25.20
0.13	24.91	23.47	22.50	25.53
0.14	25.25	23.93	22.84	25.72
0.15	25.53	24.02	23.29	27.12
0.16	25.90	24.30	23.42	28.70
0.2	35.66	34.58	35.87	35.99
0.3	58.67	54.85	57.90	52.01
0.5	68.33	68.95	68.41	68.11
0.6	67.07	66.91	66.02	65.61
0.7	71.11	70.92	70.77	70.89

## Data Availability

Data sharing not applicable.

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
