# Peer review of "Co-Training for Visual Object Recognition Based on Self-Supervised Models Using a Cross-Entropy Regularization"

_entropy, 2021, doi:10.3390/e23040423_

Round 1

Reviewer 1 Report

This paper suggests a new co-training technique and applies it to visual object recognition.  I like the idea of co-training, but I'm concerned about how this co-training procedure was set up.

I have one major issue with this paper, which is that roughly, it seems to me (perhaps naively) that co-training works because the separate classifiers are each other's adversarial networks.  But in this paper, the authors throw out images that are labeled differently by the two networks.

If the authors change that, I am happy to reconsider rejection.

A few minor comments:

-- "learnin" to "learning"

-- Line 23, "The great difficulty..." seems to me to be true of a single object as well as a class of objects, e.g. bispectrum

-- Line 34, "Deep learning models for visual recognition..." seems to me to be an overstatement, e.g. early stopping

-- Line 58, This is not how I think about regularization, as it has nothing to do with limited data

-- Could be clearer about what "pseudo-labels" are

-- I'm not sure I see that different views are represented by the different networks in the training algorithm, so more care should be taken to prove that in experiments these different views really are represented

-- In pseudo-code, no need for an & in the 6th to last line given the if statement above, and switch Add and Delete in the line below.

Author Response

(see the attached file)

Reviewer 2 Report

The paper describes a training strategy utilizing unlabeled data in a supervised learning task. For evaluation the task of visual object recognition is used. The claimed novelty lies in the introduction of a cross-entropy regularization term, that is used to control the inclusion of unlabeled data into the training set.

Overall the paper is well written, although it is in some points a little bit unclear, which makes requires to jump forth and back to understand the discussed topics. In total the paper suffers from several significant drawbacks:

  • At some points the terminology is a little bit confusing. A more clear-cut distinction between self-supervised and semi-supervised components what have been helpful to follow the argumentations.
  • The reference are all in all a little bit outdated. The related work section leaves out several aspects that should have been discussed. These are e.g. the relation between the co-adaptation approach and methods for auto-augmentation. Further, the topic of data augmentation has not been discussed.
  • There is a huge bunch on very recent literature on self-supervised learning including many different strategies to define the learning goal on self-supervision:

    @article{ title={Self-supervised Visual Feature Learning with Deep Neural Networks: A Survey},
     author={Longlong Jing and Yingli Tian},
      journal={IEEE transactions on pattern analysis and machine intelligence},
      year={2020} }

    This makes it hard to understand why only two very simplistic self-supervised strategies are used here. A more in depth evaluation of different self-supervision strategies would have been helpful.

  • The illustration would benefit from separating architectural elements from the overall information flow. At least an indicator what function is calculated in the network blocks would be helpful
  • Overall, the evaluations suffer from two significant drawbacks:
    • The used datasets are nowadays used for proof-of-concepts and illustration purposes. To validate the proposed approach datasets like OpenImagesV6 could be used. It is very straightforward to cut out bounding box annotations an resize the them in order to generate a large scale, high quality state-of-the-art dataset for object recognition.
    • The selection of competing approaches is unclear. Further, a discussion on the functionality and differences of the competing approach is missing in the related work section.

Author Response

(see attached file)

Reviewer 3 Report

This paper describes an image classification system based on the co-training principle.  The authors propose an original way to build multiple "views" of the input image and use an augmented loss function for the model training. The paper is well written and technically sound. However, the experimental part is not convincing and needs some more details and experimental results.  Especially concerning are tables 2 and 3 where the comparison is partly not fair or provided numbers are cherry-picked.

Comments and suggestions:

  1.  Line 32: "... unsupervised learning [4] is based on creating a model that allows finding a label of an input data considering that the labels of the images are not accessible". This statement is misleading since a label can't be found if it's not available. Unsupervised learning allows finding characteristics of the data which can be assigned labels if necessary.
  2. Line 43: "... monitored data[7]." Here, "monitored" is not appropriate. Either explain what "monitored" means in terms of co-training or replace it.
  3. Line 118: "... noisy data." -> "... noisy labels."
  4. Algorithm 1:  "Accuracy" with respect to a single image is either 0 or 100%. So, comparing it with a threshold doesn't make sense. Probably, the authors mean output "Probability" or "Score". Also, "Xu.Add(Image_i)" -> "Xl.Add(Image_i)" and "Xl.Delete(Image_i)" -> "Xu.Delete(Image_i)".
  5. Reference [18]. The authors cite it several times as a closely related work from which they borrowed the data partitioning scheme (Line 248), results (Table 2), or as a possible way to extend the number of views (Line 228). Actually, reference [18] describes quite a different task, i.e. depth prediction, and uses totally different data. Unless it's the authors' mistake, the relevance of this reference is at least questionable.
  6. MNIST database. How many labeled images were used from this database?
  7. Lines 260, 261: This is the only sentence where a "validation sample" (?!) is mentioned. Much more details about the validation data are needed such as where they came from; were they the same for each database or different; how were they selected; which hyper-parameters were tuned using them, etc.
  8. Table  2: Where the results of Deep Co-Training[18] come from?  The result of Co-Teaching[6] is obtained from a single view. It can be argued whether it's fair to compare results from a different number of views, but at least this difference should be mentioned. The results of Mean Teacher[3] are for the case when the ConvNet is used. For the ResNet  (used by the authors) the results are better.
  9. Table 3. There are also differences in the experimental conditions of the different methods similar to those from Table 2.
  10. Line 317: "... Tables 2 y 3." -> "... Tables 2 and 3."
  11. Most of the studies cited in this paper present their results as a mean and std of 4 to 10 runs.  To make the comparison fair in Tables 2 and 3, authors should also show the performance of their system in the same way.

Author Response

(see attached file)

Round 2

Reviewer 1 Report

This paper seems improved, but I'm confused on some technical details related to your test accuracy.  Your algorithm in pseudocode seems to have as its endpoint a better labeled dataset, throwing out controversial data-- but that is not your stated goal.  It might make sense to explain in pseudocode how the two networks combine to give one prediction for the label when the two networks disagree.  The reason for this-- it currently sounds like when you report test accuracy, that you might be throwing out data that the networks disagree on.  I suspect it's just an ensemble average, but I'm not sure.  Apologies for the pointed comment.  As soon as I understand this, I'm happy to recommend acceptance.

Author Response

(see attached file)

Reviewer 2 Report

Overall most of my concerns have been approprietly addressed by the authors. In addition, i really appreciate the sincerity of the authors regarding given some of the limitations. The article can be accept on the present form, despite some minor typos that should be corrected.

Author Response

(see the attached file)

Reviewer 3 Report

After the revision, the paper has become clearer and easier to read. Most of the missing experimental details are added and descriptions are fixed. I think there are no more big flows preventing the paper from being published.

Author Response

(see the attached file)
